# Religiosity Is Associated with Reduced Risk of All-Cause and Coronary Heart Disease Mortality among Jewish Men

**DOI:** 10.3390/ijerph191912607

**Published:** 2022-10-02

**Authors:** Sigal Eilat-Adar, Devora Hellerstein, Uri Goldbourt

**Affiliations:** 1Healthy and Active Lifestyle Education, Academic College at Wingate, Netanya 4290200, Israel; 2Sackler Faculty of Medicine, School of Public Health, Tel Aviv University, Tel Aviv 6997801, Israel; 3School of Education, Academic College at Wingate, Netanya 4290200, Israel

**Keywords:** religious practice, coronary heart disease, mortality, morbidity, epidemiology

## Abstract

Previous studies have found an inverse association between religiosity and mortality. However, most of these studies were carried out with Christian participants. This longitudinal study aimed to determine whether a composite variable based on self-reported religious education and religious practices is associated with coronary heart disease (CHD) and all-cause mortality in 9237 Jewish men aged 40–65 years at baseline, over a 32-year follow-up. Jewish men were characterized by their degree of religiosity, from the Ultra-Orthodox (“Haredim”)—the strictest observers of the Jewish religious rules, and in descending order: religious, traditional, secular, and agnostic. Demographic and physical assessments were made in 1963 with a 32-year follow-up. The results indicate that Haredim participants, in comparison to the agnostic participants, had lower CHD mortality. Hazard ratio (HR) and 95% confidence interval (95% CI)—adjusted by age, cigarette smoking, systolic blood pressure, diabetes, socioeconomic status, BMI, and cholesterol, was: [HR = 0.68 (95% CI 0.58,0.80)] for Haredim; [HR = 0.82 (95% CI 0.69,0.96)] for religious; [HR = 0.85 (95% CI 0.73–1.00)] for traditional; and [HR = 0.92 (95% CI 0.79–01.06) for secular, respectively (*p* for trend = 0.001). The same pattern was observed for total mortality. This study shows an association between religious practice among men and a decreased rate of CHD and total mortality.

## 1. Introduction

Orthodox Jewish men lead a lifestyle which differs in many aspects from that of their non-religious peers. They follow strict dietary rules (‘Kashrut’), exhibit a life-long preference for scholarly activity rather than physical activity, and may under-use medical services [1]. This pattern has persisted over the generations.

It would seem that some aspects of this lifestyle could be detrimental to their health and longevity. However, previous studies have found an inverse association between religiosity and mortality [2,3]. Most of these studies were carried out with Christian participants [4]. In a meta-analysis of 69 studies (28 articles) carried out in healthy populations, religiosity/spirituality was associated with reduced mortality [combined hazard ratio (HR) = 0.82, 95% confidence interval (95% CI) 0.76, 0.87, *p* < 0.001]. This association was independent of behavioral and lifestyle factors (smoking, drinking, exercising, socioeconomic status, and social support) [5]. In an observational study of 92,395 participants from the Women’s Health Initiative, religious affiliation was inversely associated with total mortality [adjusted HR = 0.84 (95% CI 0.75,0.93)] [6].

In a study with Jewish participants who were predominantly American, trust in God and religious observance were associated with lower levels of depressive symptoms compared to the levels found in those with mistrust in God [7]. However, in a review on religion and spirituality, the results were conflicting. While the findings of some studies indicate that religiosity may promote mental health, findings from other studies indicate that religious observance might turn religion from a potential resource into a source of spiritual struggle—especially among psychiatric patients—by means of negative religious coping (such as miscommunication, interpersonal or negative encounters with other believers, internal religious guilt, doubt, or anger with God), as well as misunderstandings (especially regarding health care services) [8].

In the cross-sectional Study of Health, Aging, and Retirement in Europe (SHARE—Israel survey) [9], synagogue attendance was a significant predictor of better health in six of seven health measures among 1287 adults 50 years old or over. Prayer, by contrast, was found to be inversely associated with health, as measured in self-rated health, long-term health problems, and activity limitation, as well as in validated measures of diagnosed chronic diseases, physical symptoms, and activities of daily living. This inverse association might reflect the use of prayer as a coping mechanism for individuals with health problems.

As reported in the Health and Retirement Study [10], 18,370 American participants aged 50 and older were interviewed in 2004 and tracked for all-cause mortality until 2014. The risk of total mortality was compared between participants who reported being religious and those who reported having “no religion”. Participants reporting religion as “very important” compared to others, had a mortality risk of [HR = 1.13 (95% CI 1.09, 1.16)]. Jews, who constituted 2.45% of the participants, exhibited a lower mortality risk [HR = 1.65 (95% CI 1.34, 2.03)] compared with those who declared having “no religion”. Further adjustment for socioeconomic status (SES) eliminated the difference [HR = 0.92 (95% CI 0.75, 1.12)].

Coronary Heart Disease (CHD) is the leading cause of death in much of the Western world. In the USA, it accounts for one-third of the deaths of adults above the age of 35 [11]. In Israel, it is the second leading cause of mortality following cancer, accounting for 15% of all deaths. CHD affects 10.2% of Israeli men [12]. Few studies have assessed the association between religious belief and CHD morbidity and mortality. In the Women’s Health Initiative described above, while religious affiliation was associated with a lower risk of all-cause mortality, it was not associated with a reduced risk of coronary heart disease (CHD) morbidity or mortality [5]. However, in a meta-analysis of observational studies, Islamic religion/spirituality was assessed by a self-administered 102-item questionnaire. Participants in the top quartile of religion/spirituality exhibited decreased odds of having CHD compared to participants in the lowest quartile [odds ratio (OR) = 0.20 (95% CI 0.06,0.59)] [6].

Based on the scant and conflicting evidence, the current study aimed to examine whether there is an association between religiosity among Jewish men and a risk of CHD and mortality. For this purpose, the researchers examined whether a composite variable based on reported religious education and reported religious practice is associated with all-cause and CHD mortality rates in Jewish Israeli men aged 40–65 years at baseline over a 32-year follow-up, after adjustment for known confounders.

## 2. Methods

### 2.1. The Subjects

The current study used a retrospective longitudinal research design based on the data collected by the Israeli Ischemic Heart Disease (IIHD) Project in 1963 on 11,876 Jewish civil servants and municipal employees. The original study sample was stratified by six geographical areas of origin. Inclusion criteria included males aged 40–65, working in the three largest urban areas in Israel. The participants were assessed at enrollment (in 1963) and at two follow-up visits (in 1965 and 1968). Subsequent to these assessment visits, CHD mortality and total mortality were recorded at 23 and 32 years of follow-up.

Out of 11,876 participants, 173 men were excluded due to an origin other than the six predefined ones, 117 men who died before the second assessment, and 227 men who did not return for assessment. Religiosity information was unavailable for 478 men. The number of individuals in the current analysis was thus reduced to 9237 men, among whom 5953 (64.4%) died over the 32 years of follow-up. Of these, 1622 (27.2%) died from CHD.

The extent of religiosity, according to belief and practice, on a scale from 1 to 5, was defined in 1965 as follows: (1) Ultra-Orthodox (Haredim) are defined as following the strictest observance of religious rules and are distinguished from the rest of the population by the following: way of dressing; adherence to all the rules of the ‘Halakha’ (Jewish Law) to the letter, including the most severe demands of Kosher food production and certification (‘Kashrut’); strict observance of the many rules related to married life and practice; rigorous adherence to all the rules related to the Sabbath (e.g., non-use of transportation or other mechanized or electric devices); and, an absolute prohibition of females and a large percentage of males from entering military service. In addition, during the 1950s and 1960s, the vast majority of Haredim also avoided voting in general and local elections; (2) The “religious” group resembles the Haredim in terms of obeying the Halakha, but differs considerably in that they make up an integral part of the Zionist Israeli society. Religious Jews played a significant role in establishing the Moshavim and Kibbutzim (the original cooperative farming communities) in pre-independence Israel, and they and their sons routinely fulfill their obligatory army service. They also adhere to kosher food regulations. Unlike the Haredim, for the most part, they do not reside in a concentrated manner in specific areas of certain towns or cities, and with the exception of wearing a “kippa” (skullcap), they dress like the general population; (3) The “traditional” group represents those who are close to religion but do not pray daily, and on the Sabbath they might travel, go to the theatre or a movie, and use electricity. Some of them attend services on Rosh Hashanah and Yom Kippur, but otherwise they do not regularly attend synagogue. Their daughters normally do not abstain from army service on claims of religious orthodoxy; (4) The “secular” do not keep kosher, disregard most limitations imposed by a strict religious interpretation, serve in the military, and engage in unlimited activities on the Sabbath. (5) Those who declared themselves to be “nonbelievers” were categorized as “agnostic”. The extent of the subjects’ religiosity was scaled from Haredim to agnostic [13].

### 2.2. Definition of Covariates

Baseline measurements were taken on the first visit during enrolment in 1963. Socioeconomic status was evaluated by a five-point index (1—lowest to 5—highest) based on education (nine levels ranging from no formal schooling to a graduate degree) and occupation (five levels ranging from “laborer” to “professional”) [14].

Smoking habits were categorized as a dichotomy variable, according to participant response to the question of if they have “ever smoked” or “never smoked”. Body mass index (BMI) was defined based on weight (to the nearest kilogram, wearing trousers only) divided by the square of height (to the nearest centimeter without shoes). Systolic blood pressure (SBP) and diastolic blood pressure were measured in the right arm with the subject in a recumbent position, both 30–45 min after arrival at the clinic and 15–30 min later. These measures were taken in 1963. Total blood cholesterol was measured in 1965 using the Anderson and Keys method [15]. A diagnosis of diabetes was based on serum glucose levels (collected for all subjects in the IIHD study) with an oral glucose tolerance test when indicated, on the use of oral hypoglycemic/insulin therapy, or a clinical history of diabetes confirmed by the primary care physician. A man diagnosed with diabetes up to 1968 was classified as diabetic. CHD and all-cause mortality were recorded at 23 and 32 years of follow-up.

### 2.3. Statistical Analysis

The researchers estimated the hazard of CHD, as well as total mortality associated with the different categories of self-reported religious denomination and practice, relative to the agnostic one, using the Cox proportional hazard method.

A multivariate analysis was adjusted for age, smoking, SBP, diabetes, BMI, and SES. The *p* for trend was tested by the Mantel–Cox test. Stata version 15.1(STATA, version 15; StataCorp LP, College Station, TX, USA) was used for data analyses [16].

## 3. Results

The Haredim were 1.4 years older, comprised of fewer smokers, had a higher percentage of diabetes prevalence, slightly lower height and BMI, and lower SES. A lower rate of CHD mortality and a trend of lower all-cause mortality were also recorded for this group (See Table 1).

Degree of religiosity was related to SES (See Table 2).

The findings indicate that the higher the SES, the lower the religiosity. At 32 years of follow-up, it was found that the degree of religiosity was inversely related to CHD mortality and all-cause mortality.

HR and 95% CI are presented in Table 3. Haredim, religious, and traditional Jews had a significantly lower risk for CHD as well as for all-cause mortality, with a significant trend of lower rates as the degree of religiosity was higher. These trends were not appreciably altered after controlling for sociodemographic and anthropometric confounders.

## 4. Discussion

The present study shows that both CHD and all-cause mortality are negatively associated with an increasing degree of religiosity, and that the men with the strictest observance had the lowest mortality rate over the 32 years. These associations were not appreciably altered after controlling for sociodemographic or anthropometric confounders. In the current study, although baseline SES was associated with higher mortality, controlling for it did not alter the strong positive association of religiosity and a lower risk of CHD or of all-cause mortality.

The current study found that Haredi Jews had the lowest SES status. While lower SES is usually associated with higher CHD and all-cause mortality [17,18], an opposite association was found in the current study. Religious practice and lifestyle may explain the reason for the lack of effect of SES. Haredi Jews typically assemble for communal prayer three times daily, and gather regularly for communal events, religious holidays, life-cycle occasions, and religious study [19]. In addition, they are composed of relatively large families—an average of 7.1 children per ultra-Orthodox woman compared to 3.1 in the general population, as reported in 2018 [20]. These rates have remained relatively stable over the years. Fertility rates in the early 1980s were 2.87 among the general population of Jewish women compared with 6.05 among ultra-Orthodox women [20,21]. Previous studies have found that individuals from low SES who live in affiliated neighborhoods have a lower mortality rate compared to those living in unaffiliated areas [22]. It is, therefore, possible that the social support provided by a close-knit community and family life contributes to positive health effects, countering the negative health effects typically associated with a low SES, as has been argued by others [23].

CHD risk factors include modifiable risk factors associated with health behaviors, including BMI, smoking, adherence to medication treatment [24], and stress [25]. It is possible that a religious lifestyle influences some of these. In several studies conducted in the USA, a high frequency of attendance at religious services was found to be associated with beneficial health behaviors (e.g., smoking avoidance, having a personal physician) [26], increased social integration, being provided with support, as well as improved coping with stressful life events, reduced depression, and positive emotions, leading to the health-promoting physiological effects of less chronic inflammation [27,28,29]. The low prevalence of smoking in the Haredim group in our study is in line with these results. A possible explanation is that these men abstain from smoking during the Sabbath—that is, from sundown Friday to sundown Saturday, due to the religious prohibition [30]. In addition, religiosity can promote mental health and reduce stress through community support, positive beliefs (such as hope, a sense of control, comfort, and meaning), and positive religious observance (such as positive religious reframing of stressors, spiritual connectedness, and spiritual support) [8].

Our results have recently been supported by a study conducted on 230,636 Jewish Israelis (62,674 Haredim) between 1996 and 2016 [31]. After adjusting for sex, age, marital status, number of children, education, and SES, a higher mortality rate was observed among non-Haredim compared to Haredim (HR = 1.60; 99% CI = 1.52–1.68). In the current study, the researchers show that this association may be mediated through lower CHD mortality in this group.

The strengths of our study are the relatively large sample size and the longitudinal design. This study also offers an abundance of data regarding midlife health risk factors. The main limitation of this study is the lack of data on women and on people living in kibbutzim or agricultural villages; the external validity of the current study may be limited since the sample represents people who were salaried employees. Since the late 1960s, a process of decreasing smoking has taken place in Israel. In addition, during these years, the availability of new medications has significantly transformed medical treatment. The extent to which these changes may have affected the study participants was not examined.

## 5. Conclusions

The researchers identified an association between baseline levels of self-reported religiosity and a decreased rate of CHD mortality, as well as all-cause mortality, in employed middle-aged Israeli men who were followed up prospectively for 32 years, despite the higher prevalence of risk factors such as low SES and diabetes. This study adds to the scant research that has been conducted on the specific population of Jewish men. It demonstrates the significance of religiosity as expressed by belief and practice for longevity. Although several studies have been conducted among Christian populations, this research may inspire similar studies among adherents of other religious communities. The results may help in planning future studies assessing quality of life and morbidity in this population, based on health behavior changes over time. In addition, the results can contribute to future interventions to enhance SES and reduce diabetes among the specific Haredi population. At the same time, as religiosity may play a role in longevity, spirituality may offer benefits to the non-religious population.

## Figures and Tables

**Table 1 ijerph-19-12607-t001:** Sciodemographic and anthropometric characteristics according to Jewish Orthodoxy (*n* = 9237).

Religiosity Classification	Haredim (*n* = 2101)	Religious (*n* = 1525)	Traditional (*n* = 1780)	Secular (*n* = 2084)	Agnostic (*n* = 1747)	*p* for Trend
Age (years) (SD)	50.2 (6.9)	48.8 (6.6)	48.4 (6.7)	48.8(6.6)	49.5 (6.8)	<0.001
Ever smoked ^a^ *n* (%)	1235 (58.7)	1033 (67.6)	1259 (70.7)	1499 (71.9)	1267 (72.6)	<0.001
Systolic blood pressure (mmHg) (SD)	139 (21)	138 (21)	138 (23)	137 (21)	138 (21)	0.05
Diabetes (Yes) *n* (%)	202 (9.9)	165 (11.1)	144 (8.3)	178 (8.8)	111 (6.5)	<0.001
SES Mean ^a^ (SD)	2.2 (1.3)	2.3 (1.1)	2.7 (1.1)	2.7 (1.2)	3.1 (1.4)	<0.001
BMI kg/m^2^ (SD)	25.8 (3.6)	26.2 (3.4)	26.1 (3.2)	25.7 (3.2)	25.6 (3.0)	<0.001
Cholesterol (mg/100 mL) (SD)	200 (37)	208 (41)	206 (37)	211 (38)	214 (39)	<0.001
Number of deaths *n* (% of category)	1379(65.7)	974(63.9)	1093(61.4)	1321(63.4)	1186 (67.9)	<0.001
Number of deaths from CHD *n* (% of category)	314 (14.9)	262 (17.2)	307 (17.2)	380 (18.2)	359 (17.1)	

Abbreviations: SD—standard deviation; SES—socioeconomic status; BMI—body mass index. ^a^ Ranges of the participant characteristics were as follows: Smoking habits were categorized as a dichotomy variable—"ever smoked” and “never smoked”. SES is represented by a five-point index based on education (nine levels ranging from no formal schooling to a graduate degree) and occupation (five levels ranging from “laborer” to “professional”).

**Table 2 ijerph-19-12607-t002:** SES by five categories of religiosity [*n* (%)].

SES	Haredim (*n* = 2089)	Religious (*n* = 1518)	Traditional (*n* = 1770)	Secular (*n* = 2075)	Agnostic (*n* = 1738)	Total
1	859 (41.1)	466 (31)	312 (17.6)	366 (17.6)	197 (11.3)	2200 (23.9)
2	428 (20.5)	407 (26.0)	470 (26.6)	518 (25.0)	333 (19.2)	2156 (23.5)
3	505 (24.2)	493 (32.4)	638 (36.0)	716 (34.5)	598 (34.4)	2950 (32.1)
4	136 (6.5)	86 (5.7)	205 (11.6)	254 (12.2)	289 (16.6)	970 (10.6)
5	161 (7.7)	66 (4.4)	145 (8.2)	221(10.7)	321 (18.5)	914 (10.0)

Abbreviations: SES—socioeconomic status.

**Table 3 ijerph-19-12607-t003:** HR, 95% CI for CHD mortality and all-cause mortality by degree of religiosity.

Religiosity	Haredim (*n* = 2102)			Religious (*n* = 1527)			Traditional (*n* = 1780)			Secular (*n* = 2085)			Agnostic (*n* = 1746) Ref
	HR	95% CI	*p*	HR	95% CI	*p*	HR	95% CI		HR	95% CI	*p*	*p* for Trend
All-cause mortality ^a^	0.83	0.77, 0.90	<0.0001	0.94	0.86, 1.01	0.13	0.93	0.86, 1.01	0.08	0.95	0.88, 1.03	0.20	0.0024
All-cause mortality ^b^	0.80	0.73, 0.87	<0.0001	0.86	0.79, 0.94	0.001	0.88	0.81, 0.95	0.002	0.92	0.85, 0.99	0.03	0.0001
Coronary heart disease mortality ^a^	0.63	0.54, 0.73	<0.0001	0.84	0.72, 0.99	0.03	0.87	0.75, 1.01	0.07	0.91	0.79, 1.05	0.21	0.001
Coronary heart disease mortality ^b^	0.68	0.58, 0.80	<0.0001	0.82	0.69, 0.96	0.02	0.85	0.73, 1.00	<0.05	0.92	0.79, 1.06	0.24	<0.001
All-cause mortality ^a^	0.83	0.77, 0.90	<0.0001	0.94	0.86, 1.01	0.13	0.93	0.86, 1.01	0.08	0.95	0.88, 1.03	0.20	0.0024

^a^ Adjusted for age. ^b^ Adjusted for age, cigarette smoking, systolic blood pressure, diabetes, socioeconomic status, body mass index, and cholesterol.

## Data Availability

For ethical reasons and protection of confidentiality, data from the IIHD Project can be provided upon request made to the leading author and approval by the institutional ethics board.

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
