# Peer review of "Religiosity Is Associated with Reduced Risk of All-Cause and Coronary Heart Disease Mortality among Jewish Men"

_ijerph, 2022, doi:10.3390/ijerph191912607_

Round 1
Reviewer 1 Report
The manuscript premise is very interesting and certainly contributes to the lack of evidence found in the literature specific to the Jewish population and specifically to CHD.
One comment is that the authors mention that the objective was to examine whether the association between religiosity and mortality is mediated by CHD. However, mediation analyses was not conducted. Either the authors should remove the statement and modify it to statement simply they are interested in religiosity effect on both mortality and CHD risk or they should examine separate models with and without CHD to determine mediation %.
Last comment, is there information on CHD related death? if there is, this would be nice outcome to add.
Author Response
Reply to the reviewers:
We wish to thank you for the positive feedback and for the insightful comments which we hope to have appropriately addressed.
Below each comment is our response highlighted in yellow.
In the Word Document we have also highlighted in yellow the changes we have made.
The manuscript premise is very interesting and certainly contributes to the lack of evidence found in the literature specific to the Jewish population and specifically to CHD.
One comment is that the authors mention that the objective was to examine whether the association between religiosity and mortality is mediated by CHD. However, mediation analyses was not conducted. Either the authors should remove the statement and modify it to statement simply they are interested in religiosity effect on both mortality and CHD risk or they should examine separate models with and without CHD to determine mediation %.
Thank you for clarifying to us the ambiguity in the objective. We have changed the phrasing of the so that it is more focused:
… the current study aimed to examine whether there is an association between religiosity among Jewish men and a risk of CHD and mortality. (Lines 77-79)
Last comment, is there information on CHD related death? if there is, this would be nice outcome to add.
The data is presented in Table1 and 3. We thank you for highlighting to us that it was not clear, and have changed 'risk of CHD and all-causes mortality' to 'risk of CHD mortality and all-cause mortality' to differentiate from 'risk of CHD morbidity'.
A lower rate of CHD mortality and a trend of lower all-cause mortality were also recorded for this group. (Lines 155-156)
…At 32 years of follow-up, it was found that degree of religiosity was inversely related to CHD mortality and all-cause mortality. (Lines 162-163)
Table 3. HR, 95% CI for CHD mortality and all-cause mortality by degree of religiosity.
Reviewer 2 Report
This study represents valuable research, but unfortunately not in a very precise manner. The manuscript can be published, but only with some changes. Its value is manifested by the fact that it touches on an important aspect that demonstrates how important religion is, even in the sense that is most important to today's man, namely the length of his life. The authors are clear and focused in the introduction, dedicated to the goals and why the research was necessary. They also describe their methods, and the results are consistent with the methodology and objectives. The literature is current and extensive. The discussion evaluates the results critically, but the conclusion could be expanded with clearer recommendations and limitations for future research based on this work.
The aim of the research was to show whether religiosity contributes to reducing the risk of cardiovascular diseases in Jewish men and whether it affects total mortality.
consider the topic both original and relevant because it demonstrates the significance of religion as such on a firm scientific basis in the context of the particular Jewish population (Jewish men) that is included in this research. Due to its potential to aid in the prevention of cardiovascular diseases (and to affect total mortality rate) in a particular population, this research has the potential to inspire similar research among adherents of other religious communities.
Similar studies were done on Christians. This is the initial study of its kind on this particular population.
The research methodology is well elaborated and described in detail.
The conclusion could be expanded with clearer recommendations and limitations for future research based on this work.
Author Response
Thank you for the reviewer's feedback and for pointing out the need to expand upon the conclusion and recommendation. We have included the ideas suggested by the reviewer and have expanded the conclusion section including recommendations for future research and interventions. (Lines 229-238)
This study adds to the scant research that has been conducted on the specific population of Jewish men. It demonstrates the significance of religiosity as expressed by belief and practice for longevity. Although several studies have been conducted among Christian populations, this research may inspire similar studies among adherents of other religious communities. The results may help in planning future studies assessing quality of life and morbidity in this population, based on health behavior changes over time. In addition, the results can contribute to future interventions to enhance SES and reduce diabetes among the specific Haredi population. At the same time, as religiosity may play a role in longevity, spirituality may offer benefits to the non-religious population.
Limitations are presented at the end of the discussion section (Lines 218-224)
The main limitation of this study is the lack of data on women and on people living in kibbutzim or agricultural villages; the external validity of the current study may be limited, since the sample represents people who were salaried employees. Since the late 1960s, a process of decreased smoking has taken place in Israel. In addition, during these years the availability of new medications has significantly transformed medical treatment. The extent to which these changes may have affected the study participants was not examined.
Reviewer 3 Report
â–ºIt is my honor & pleasure having the opportunity to review your research work. It is both important & timely topic. I applaud researcher’s efforts presenting these important findings. However, the following are some recommendations that must be followed by researchers.
Line 48, “regarding medical treatment”
I would suggest avoid using biased language. Therefore, I recommend using health care services instead.
Line 66 “CHD”
Start your sentence with the full term. Like the following:
Coronary Heart Disease (CHD).
Line 77. I would highly recommend adding the research main question at the end of this section.
Line 79 “For this purpose, we examined”
Please avoid using the pronoun we. It is not preferred in academic writing. Instead use researchers.
Line 84-98
Should be removed. These line are not a part of the main manuscript.
Line 99. I would definitely recommend starting the methods section with a detailed paragraph, presenting the used research design. Which is “Retrospective longitudinal”. Of equal importance, show your reader that this design aims basically at proving causality.
Line 277. I would recommend adding the flavor of theoretical literature in explaining the highlighted phenomenon.
Line 248. “These results may help in planning future interventions in this specific population.” Please elaborate more by adding how the findings of this study may be reflected in terms of helping planning future interventions in this specific population. In other words, give tangible examples that clarify this to your reader.
Author Response
We wish to thank you for the positive feedback and for the insightful comments which we hope to have appropriately addressed. Below each comment is our response highlighted in yellow.
In the Word Document we have also highlighted in yellow the changes we have made.
Reviewer 3
It is my honor & pleasure having the opportunity to review your research work. It is both important & timely topic. I applaud researcher’s efforts presenting these important findings. However, the following are some recommendations that must be followed by researchers.
Thank you very much for the positive and supportive feedback. It is our honor and pleasure to receive the reviewer's feedback which is much appreciated.
Line 48, “regarding medical treatment” I would suggest avoid using biased language. Therefore, I recommend using health care services instead.
Thank you for this comments. It has been changed in the manuscript accordingly.
Line 66 “CHD”
Start your sentence with the full term. Like the following:
Coronary Heart Disease (CHD).
This has been added according to the comment.
Line 77. I would highly recommend adding the research main question at the end of this section.
We are grateful for this suggestion, as we understand the objective was ambiguous. We have rephrased the aim in order to clarify it for the reader:
… the current study aimed to examine whether there is an association between religiosity among Jewish men and a risk of CHD and mortality. (Lines 77-79)
Line 79 “For this purpose, we examined”
Please avoid using the pronoun we. It is not preferred in academic writing. Instead use researchers.
Thank you for the comment. This change has been made here and throughout the text. (Lines 79, 142, 214, 226)
Line 84-98
Should be removed. These line are not a part of the main manuscript.
In order to define the study sample, we believe it is important for the reader to understand the source of the database. But, we understand from this comment that the description is over-detailed, and the section has been made more concise.
Line 99. I would definitely recommend starting the methods section with a detailed paragraph, presenting the used research design. Which is “Retrospective longitudinal”. Of equal importance, show your reader that this design aims basically at proving causality.
The study design was added. We are cautious about using the term 'causality' in this type of study, and therefore, have throughout used the term 'association'
Line 277. I would recommend adding the flavor of theoretical literature in explaining the highlighted phenomenon.
As the manuscript ends in line 238, we understand the reviewer meant a different line. We searched and were not certain to which phenomenon he/she referred. We will be happy to address this if the line intended is sent to us.
Line 248. “These results may help in planning future interventions in this specific population.” Please elaborate more by adding how the findings of this study may be reflected in terms of helping planning future interventions in this specific population. In other words, give tangible examples that clarify this to your reader.
We assume the line referred is 238. Thank you for this highlighting the need to expand upon the conclusions. The section below was added:
This study adds to the scant research that has been conducted on the specific population of Jewish men. It demonstrates the significance of religiosity as expressed by belief and practice for longevity. Although several studies have been conducted among Christian populations, this research may inspire similar studies among adherents of other religious communities. The results may help in planning future studies assessing quality of life and morbidity in this population, based on health behavior changes over time. In addition, the results can contribute to future interventions to enhance SES and reduce diabetes among the specific Haredi population. At the same time, as religiosity may play a role in longevity, spirituality may offer benefits to the non-religious population. (Lines 229-238)